# The role of government policy, social infrastructure and Fengshui in intending to buy tourism real estate

**Khan Van Ma[1], Phuong V. Nguyen[1]\*, Zafar U. Ahmed[2]**

**1** Center for Public Administration, International University, Vietnam National University, Ho Chi Minh City, Vietnam, **2** The School of Business, International University, Vietnam National University, Ho Chi Minh City, Vietnam

\* nvphuong@hcmiu.edu.vn

**Data Availability Statement:** All relevant data are within the paper and its Supporting information files.

**Funding:** The author(s) received no specific funding for this work.

## Abstract

People typically purchase residential properties for two reasons: to live in or invest. However, both purposes necessitate careful consideration before deciding because high financial costs are involved, and housing loans are typically considered necessary for this purpose. Customers' demands are constantly changing, becoming more complicated with higher requirements. The focus of this research is on tourism real estate selection. This market in Vietnam is still new and emerging and has encountered numerous issues regarding government policy, finance, and land authorization for constructing, owning, and managing. Because the form of tourism real estate is still new, customers are hesitant about investing in or buying these properties. Hence, to compete in the current fiercely real estate industry, real estate firms must understand their customers' expectations by frequently involving customer research in the company's strategy. However, there is still a lack of research on the connection between these factors and individual expectations in the well-known philosophy of the Theory of Planned Behavior (TPB), leading to behavioral intentions. Therefore, to fulfill the gap in the previous literature, this paper aims to investigate the connection between these factors with core variables of TPB, hence, addressing the current problems in the real estate industry. 471 valid respondents in Vietnam were collected for data analysis through two survey approaches. PLS-SEM was used to test hypotheses due to the relationship complication in the conceptual models. The results show that government policy influences attitudes and perceived behavioral control, whereas social infrastructure affects social norms and perceived behavioral control. Moreover, Fengshui ambient condition also positively influences all three core factors: attitudes, social norms, and perceived behavioral control. Finally, these factors impact on intention to buy tourism real estate. Through results, this paper has developed a purchase intention model through social aspects of the tourism real estate industry. In addition, this paper demonstrates the connection between social factors and individuals' expectations for a purchase intention, providing the importance of the government's role, architecture style, and social infrastructure in the marketing literature of the real estate industry. As a result, managers and governments need to take advantage of new releases of government regulations in time to enhance customers' positive attitudes toward purchasing tourism real estate. Moreover, social infrastructure and Fengshui conditions are

**Competing interests:** The authors have declared that no competing interests exist.

crucial to establishing social norms and perceived control, aiming to leverage the intention to purchase tourism real estate. Thereby, recommendations of marketing strategies based on these findings were suggested to attain the optimal result for sales. Finally, this research also includes some limitations. Hence, suggestions for further research were also provided, such as possible moderation, possible mediating effects, or control of data bias.

## 1. Introduction

People typically purchase residential properties for two reasons: to live in or invest. However, both purposes necessitate careful consideration before deciding because high financial costs are involved, and housing loans are typically considered necessary for this purpose. Government policies play an essential role in customers' decision-making process regarding buying real estate. For example, government incentives for housing purchases can increase the sale of houses [1]. Social infrastructure is another critical element affecting customers' attitudes towards housing constructions, affecting their willingness to buy those buildings. According to Vischer [2], users make judgments about constructions and other designed facilities based on personal experiences and interrelations with them, which is opposed to experts who design and build facilities and infrastructure but may never use them [3]. When purchasing real estate, buyers usually consider Fengshui, predominantly Asian customers, as another factor. According to Lin et al. [4], FengShui is an important factor for many Chinese in making choices such as where to live, where to work, where to bury their loved ones, where to place buildings, and how to design both the inside and outside of of the buildings. Fengshui is not only used in Asian and nearby countries but also in Western countries; for example, Fengshui principles have been involved in the various design of Western architecture and interior decoration projects [5].

Customer demands are constantly changing and have become more complicated with higher requirements. The focus of this research is on the tourism real estate industry. This market in Vietnam is still new and emerging and has encountered numerous issues regarding government policy, finance, and land authorization for constructing, owning, and managing. Because the form of tourism real estate is still new, customers are hesitant about investing in or buying these properties. Hence, to compete in the current fiercely tourism real estate industry, real estate firms must understand their customers' expectations by frequently involving customer research in the company's strategy.

Several papers examined the factors that affect customers' intention to buy tourism real estate, including the government policy, social infrastructure, and Fengshui conditions. However, there is still a lack of research on the connection between these factors and individual expectations in the well-known philosophy of TPB, leading to behavioral intentions. Therefore, to fill the gap in previous research, this study applied three core elements of TPB that shape an individual's behavioral intentions: attitudes, subjective norms, and perceived behavioral control [6]. As a result, this paper aims to investigate the connection among these factors with core variables of TPB, hence, addressing the current problems in the real estate industry.

## 2. Literature review

### 2.1. Government policy

Real estate is crucial to a nation's economic production and development. According to Zhu [7] since land supply is essentially fixed in terms of geography and quality, the urban land

industry is inherently flawed and may not react effectively to market changes. Consequently, there are considerable government interventions in several nations' land markets [8]. Those interventions can include direct government control over land supply or land use rules and frequently disrupt the whole marketplace. For example, in 2016, the Chinese government initiated a policy known as "reduce inventory." The strategy seeks to strengthen taxation and credit policies that support rational housing purchases, place a premium on satisfying a rigid need, and minimize housing inventory. Rural migrants are considered essential potential buyers in small cities under this program. Rural migrants benefit from reduced taxes and down payment obligations. Local authorities also promote housing purchases and make urban Hukou easily accessible [1].

In fact, according to Mohit and Azim [9], through research on residential satisfaction for new advanced public low-cost accommodations in Kuala Lumpur, Malaysia, people are assumed to investigate the physical characteristics of housing infrastructure and services depending on the current needs and priorities as well as some pre-determined standards created by government agencies, experts, and professionals. Furthermore, according to Zhang et al. [10], government policies (especially government incentives) had the greatest impact on purchase intentions, including buying real estate. This justified the case of the Vietnamese government when the 30-trillion-Vietnamese-dong Home Loan Package was launched in 2013 [11]. Since the program's inception, approximately 80% of apartment purchasers in Ho Chi Minh City, Vietnam (HCMC) have taken advantage of the package [12]. Government policy towards tourism is also essential to customers' intention to buy tourism real estate. Moreover, Nguyen et al. [13] stated that tourism real estate developments are typically subject to more legal risks than regular infrastructure projects. Therefore, the following hypotheses are proposed:

H1: Government policy positively influences attitudes

H2: Government policy positively influences social norm

H3: Government policy positively influences perceived behavioral control

H4: Government policy positively influences the intention to buy

## 2.2. Social infrastructure

According to Lee and Xue [14], targets for sustainable tourism development in relation to economic and social aspects are regularly interconnected, such as raising employment and business opportunities, strengthening community facilities/infrastructure, raising the community's living standard, and boosting goods and services. Several previous studies indicated that tourism satisfaction and behavior have been proven to be affected by the socioeconomic characteristics of a region, such as social infrastructure, transportation, tourism choices and amenities, and relative price behavior [15–17].

In addition, social infrastructure plays an important role in purchasing tourism real estate. According to Lee et al. [18], in terms of sociocultural management, infrastructure should be accommodated people with disabilities. A study of 156 residents in progressively created low-income public housing in Ogun State, Nigeria, discovered that one of the three most significant aspects of housing is the accessibility of social infrastructure [19]. Specifically, social infrastructure, including water supply, sanitation, and waste management facilities, is one of the essential factors for adequate housing. According to McCray and Weber [20], sufficient housing perception is a composite image of all aspects of the surroundings required to maintain a minimum acceptable quality of life. They said that cultural context, housing norms and values,

and experience with various home characteristics and norms, all influence these images. Therefore, this paper posits:

H5: Social infrastructure positively influences attitude

H6: Social infrastructure positively influences social norm

H7: Social infrastructure positively influences perceived behavioral control

## 2.3. Fengshui ambient conditions

According to Webster [21], Fengshui came from ancient China around five to six thousand years ago. After that, it was expanded to other Asian countries such as Korea, Vietnam, Thailand, Japan, Indonesia, and so on [22, 23]. Moreover, Mak and Ng [24] indicated that the Compass School and the Form School are Fengshui's two primary schools of thought and practice. The Form School, according to Wu et al. [25], emphasizes Fengshui depending on the assessment of the physical surroundings, such as a mountain and a river. The theory is founded on the geography of the constructions and settings. In the Fengshui design framework, Mak and Ng [24] developed four design modules: surrounding environment, external layout, internal layout and interior arrangement. Regarding the surrounding environment, there are seven criteria and 14 related fengshui factors as illustrated in S1 Table.

Luk et al. [26] stated in their model on customers' opinions of Fengshui that customers have three perspectives on Fengshui, including instrumental, spiritual, and minimalist. In terms of instrumental viewpoint, customers believe that Fengshui designs can influence their luck and fortune. According to the spiritual perspective, consumers consider Fengshui to be a component of their spiritual lives and cultural identities. Consumers regard Fengshui as pure superstition, according to the minimalist viewpoint. Furthermore, Sia et al. [27] stated that house purchasers that take an instrumental perspective on Fengshui will purchase homes with favorable Fengshui factors that will bring them the best of luck and fortune. They would avoid purchasing houses with unfavorable Fengshui elements that could bring them bad luck and tragedy. Therefore, in this study, we develop the following hypotheses:

H8: Fengshui ambient conditions positively influences attitude

H9: Fengshui ambient conditions positively influences social norm

H10: Fengshui ambient conditions positively influences perceived behavioral control

## 2.4. Theory of planned behavior

Despite the difficulty of understanding human behavior, TPB, created by Ajzen in 1985, has become one of the most widely utilized behavioral theories for judging people's intentions and behaviors [10]. The TPB framework predicts behavioral intention based on three components: attitude toward behavior, subjective norm, and perceived behavioral control [6]. Attitude is defined as an individual's positive or negative appraisal of the execution of specific behavior, considers subjective norm as a social component referring to felt social pressure to do or refrain from performing a specific behavior, and views perceived behavioral control as the perception of the difficulty level of carrying out that behavior, which is influenced by prior experiences and anticipated obstacles.

TPB has been used to predict customer intentions and behavior in a variety of contexts, including green residence [28], green goods [29], green housing [30–32], green hotels [33], and electric vehicles [34]. As a result, TPB is a valuable paradigm for defining ecologically

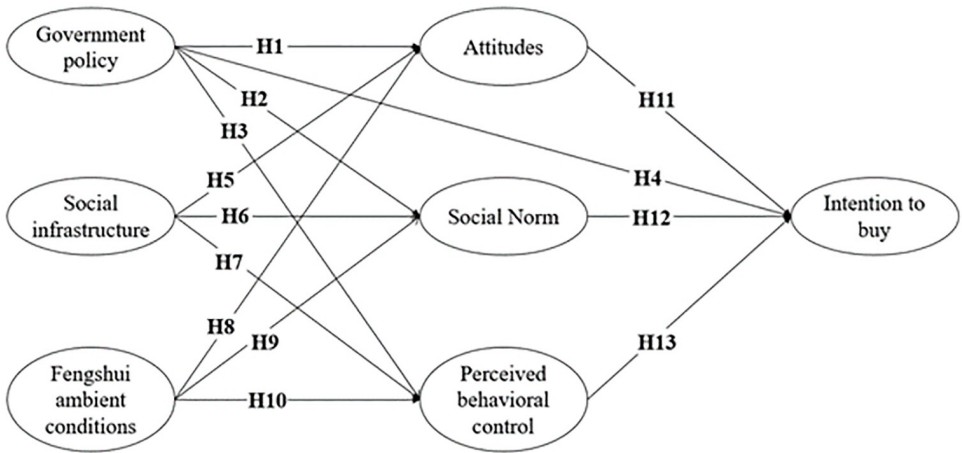

**Fig 1. The research model.**

responsible conduct [10, 33]. Furthermore, Zhang et al. [35] observed that the TPB partially favors acquiring industrial brownfield real estate. When people's perceived behavioral control is substantial, they are more likely to execute their behavioral intentions and report intentions consistent with their views and subjective norm [36]. Hence, we proposed three following hypotheses:

H11: Attitudes positively influence the intention to buy

H12: Social norm positively influences the intention to buy

H13: Perceived behavioral control positively influences the intention to buy

Fig 1 presents the research model with hypothesis development.

## 3. Methodology

### 3.1. Sample and procedures

The respondents were chosen as employed people currently living and working in HCMC and nearby provinces. The target respondent for this survey is people of working age with a certain amount of income (company employees) or retired people. The data collection was conducted both online and offline. A survey link was created for the online survey to send to respondents through a google form. The respondents were directly contacted for the offline survey and asked to complete the paper-based questionnaire. Finally, 471 valid samples were selected for data analysis.

The current study involved human participants and was reviewed and approved by the ethics committee of the Center For Public Administration, International University, Vietnam National University-Ho Chi Minh City, Vietnam. Moreover, the ethics committee ruled that no formal ethics approval was required in this study because it did not collect any medical information, there was no known risk involved, it did not intend to publish anyone's personal information, and it did not collect data from underaged respondents. This research has been performed in accordance with the Declaration of Helsinki. Informed consent for participation was obtained from respondents who participated in the survey. The respondents who participated in the survey online (using the google form) or face-to-face survey were asked to read the ethical statement posted on the top of the document (There is no compensation for

**Table 1. Sample characteristics (n = 471).**

| Dimensions | Items | Frequency | Percentage |
|---|---|---|---|
| **Gender** | Male | 285 | 60.5 |
| | Female | 186 | 39.5 |
| **Age** | 19–30 | 102 | 21.7 |
| | 30–44 | 267 | 56.7 |
| | 45–59 | 99 | 21.0 |
| | 60 and above | 3 | 0.6 |
| **Education level** | High school | 17 | 3.6 |
| | College | 38 | 8.1 |
| | University | 331 | 70.3 |
| | Graduate | 83 | 17.6 |
| | Others | 2 | 0.4 |
| **Purchase purpose** | Investment | 207 | 43.9 |
| | Leisure | 116 | 24.6 |
| | Both Investment & Leisure | 126 | 26.8 |
| | Others | 22 | 4.7 |

responding, nor is there any known risk. Participants were asked not to provide their names to ensure that all information would remain confidential. They are strictly voluntary and may refuse to participate at any time). The survey only proceeds if they agree. No data was collected from anyone under 18 years old.

Among them, there were 60.5% male and 39.5% female. Concerning the age groups, respondents, who are from 30 to below 45 years, accounted a majority of the sample with 56.7%, whereas those, who are over 60 years, only accounted 0.6%. In addition, these participants reached different levels of education. In particular, most of the respondents attained a bachelor degree, accounting for the largest portion (70.3%). Other groups include graduates (17.6%), college (8.1%), and high school (3.6%). Finally, in terms of purchasing purpose of real estate, investment and leisure are the main reason for their purchase decision with 43.9% and 24.6%, respectively. Some purchased tourism real estate for investment and personal leisure, accounting for 26.8%. Other aims consist of keeping valuable properties, presents, or leasing, accounting for a small proportion of 4.7% (see Table 1).

### 3.2. Measure development

This study used the measure of previous studies with modifications to fit the tourism real estate context. First, the Fengshui measure was adapted with minor revisions from [24]. Next, we modified four items adapted from [37] and added three new items based on the current situation in Vietnam, including "Legal policy on real estate tax", "Legal transparency of authorities and investors" and "Supervision and sanction of the authorities" the measures of social infrastructure were modified from the scales of [19]. In addition, four factors, including subjective norm, attitudes, perceived behavioral control, and the scales of [35] measured intention to buy. Each question was rated based on the 5-point Likert scale from 1 = strongly unfavorable to 5 = strongly favorable.

### 3.3. Assessment method

The data were analyzed using the reflection measurement model. Since the relationships among examined variables are complicated, the Partial Least Square Structural Equation

Modelling (PLS-SEM) was applied by using SmartPLS 3.0 software to test the research hypotheses. The sample size was sufficient to guarantee the PLS method's regressions without singularities. PLS-SEM is a good option because the model is relatively sophisticated, but there is not much well-established literature on the conceptual model [38, 39]. In addition, using PLS-SEM is unquestionably advantageous when considering the unavailability of distributional assumptions in several social science investigations [40]. Therefore, the PLS path model was employed to estimate the measurement and structural models.

## 4. Results and discussions

### 4.1. Measurements model

First, reliability and convergence validity were examined through values of factor loadings, Cronbach's alpha, and composite reliability (CR), which are all larger than 0.7 [41]. In particular, items were well-loaded in each dimension ranging from 0.754 to 0.961. For Cronbach's alpha and CR, values range between 0.905 to 0.969 and 0.934 to 0.972 for perceived behavioral control and Fengshui ambient conditions, respectively. In addition, the average variance extracted (AVE) values, used to measure the convergence validity, are acceptable, as values are higher than 0.5 ranging from 0.682 (Fengshui ambient conditions) to 0.896 (Social norm). Hence, the convergence validity of the data is ensured and the reliability is confirmed (see Table 2).

Thereafter, the square root of the AVEs was calculated for comparison with the correlations between every pair of latent variables. According to Table 3, no correlation coefficients are greater than the square root of AVEs; thus, discriminant validity is ensured, according to Fornell and Larcker's criteria [42]. Furthermore, Table 3 also shows good values of the Heterotrait-Monotrait ratios, which are all below 0.85, following the threshold from Kline [43]. Therefore, the discriminant validity of the measurements is also confirmed.

### 4.2. Structural model

The purpose of PLS-SEM is to predict, instead of evaluate, mode fit. Therefore, the coefficients of determination ($R^2$) and path coefficients are alternative indicators to predict the endogenous constructs such as attitudes, social norms, perceived behavioral control, and intention to buy with variances of 33.1%, 23.1%, 33.5%, and 65.1.4%, respectively.

Table 4 showed eleven out of thirteen hypotheses were supported after running bootstrap with 5000 subsamples. In particular, among four expected outcomes, government policy positively influences on customers' attitudes ($\beta = 0.287$; $p<0.01$), perceived behavioral control ($\beta = 0.112$; $p<0.1$), and intention to buy ($\beta = 0.064$; $p<0.1$); hence, H1, H3 and H4 were supported. Moreover, while the relationship between social infrastructure and attitudes was not

**Table 2. Reliability and convergent validity.**

|  | Cronbach's Alpha (> 0.7) | rho_A (> 0.7) | CR (> 0.7) | AVE (> 0.5) |
|---|---|---|---|---|
| **Attitudes** | 0.950 | 0.951 | 0.964 | 0.870 |
| **Perceived behavioral control** | 0.905 | 0.907 | 0.934 | 0.778 |
| **Fengshui ambient conditions** | 0.969 | 0.970 | 0.972 | 0.682 |
| **Government Policy** | 0.956 | 0.962 | 0.964 | 0.792 |
| **Intention to purchase** | 0.917 | 0.918 | 0.938 | 0.751 |
| **Social infrastructure** | 0.932 | 0.940 | 0.952 | 0.832 |
| **Social norm** | 0.942 | 0.942 | 0.963 | 0.896 |

**Table 3. Discriminant validity.**

| | Fornell-Larcker Criterion | | | | | | |
|---|---|---|---|---|---|---|---|
| | 1 | 2 | 3 | 4 | 5 | 6 | 7 |
| **Attitudes** | *0.933* | | | | | | |
| **Behavioral control** | 0.651 | *0.882* | | | | | |
| **Fengshui ambient conditions** | 0.513 | 0.544 | *0.826* | | | | |
| **Government Policy** | 0.488 | 0.403 | 0.509 | *0.890* | | | |
| **Intention to purchase** | 0.669 | 0.559 | 0.510 | 0.391 | *0.867* | | |
| **Social infrastructure** | 0.476 | 0.535 | 0.761 | 0.565 | 0.513 | *0.912* | |
| **Social norm** | 0.643 | 0.534 | 0.447 | 0.327 | 0.770 | 0.456 | *0.947* |
| | Heterotrait-Monotrait Ratio (HTMT) (< 0.85) | | | | | | |
| **1. Attitudes** | | | | | | | |
| **2. Behavioral control** | 0.702 | | | | | | |
| **3. Fengshui ambient conditions** | 0.527 | 0.576 | | | | | |
| **4. Government Policy** | 0.508 | 0.431 | 0.528 | | | | |
| **5. Intention to purchase** | 0.715 | 0.614 | 0.536 | 0.413 | | | |
| **6. Social infrastructure** | 0.502 | 0.581 | 0.802 | 0.600 | 0.553 | | |
| **7. Social norm** | 0.680 | 0.578 | 0.460 | 0.341 | 0.828 | 0.482 | |

significant, the positive effects of social infrastructure on the social norm (β = 0.243; p<0.05) and perceived behavioral control (β = .241; p<0.05) were confirmed. Thus, H6 and H7 were supported. The positive impacts of Fengshui ambient conditions on attitudes, social norms, and perceived behavioral control are all significant with path coefficients of 0.304, 0.222, and 0.303, respectively; hence, H8, H9, and H10 were supported. Finally, three factors of TPB can predict customers' intention to buy real estate; therefore, H11, H12 and H13 were supported.

**Table 4. The hypothesis testing results.**

| Hypotheses | Estimate | S.D. | Results |
|---|---|---|---|
| **H1:** Government Policy -> Attitudes | 0.287*** | 0.054 | Supported |
| **H2:** Government Policy -> Social norm | 0.077 | 0.062 | Not Supported |
| **H3:** Government Policy -> Perceived behavioral control | 0.112* | 0.062 | Supported |
| **H4:** Government Policy -> Intention to buy | 0.064* | 0.034 | Supported |
| **H5:** Social infrastructure -> Attitudes | 0.083 | 0.069 | Not Supported |
| **H6:** Social infrastructure -> Social norm | 0.243** | 0.078 | Supported |
| **H7:** Social infrastructure -> Perceived behavioral control | 0.241** | 0.072 | Supported |
| **H8:** Fengshui ambient conditions -> Attitudes | 0.304*** | 0.058 | Supported |
| **H9:** Fengshui ambient conditions -> Social norm | 0.222** | 0.069 | Supported |
| **H10:** Fengshui ambient conditions -> Perceived behavioral control | 0.303*** | 0.063 | Supported |
| **H11:** Attitudes -> Intention to buy | 0.218*** | 0.055 | Supported |
| **H12:** Social norm -> Intention to buy | 0.561*** | 0.045 | Supported |
| **H13:** Perceived behavioral control -> Intention to buy | 0.091* | 0.049 | Supported |

Notes:

*p<0.1,

**p<0.05,

***p<0.01.

## 5. Conclusion and contributions

Through the data analysis results, this paper has successfully developed a purchase intention model through social aspects such as government policy, social infrastructure, and Fengshui ambient conditions toward the Vietnam real estate market in particular and the emerging market in general. Specifically, government policy moderately influences customers' attitudes toward real estate purchases. Moreover, social infrastructure has similar effects on both social norms and perceived behavioral control. Especially, Fengshui ambient conditions positively impacts all three outcome factors. Finally, attitudes have a weaker effect on intention to buy than social norms.

### 5.1. Theoretical contributions

This paper has successfully developed a model of tourism real estate purchase intention in the emerging market. First, the significant effects of government policy, social infrastructure, and Fengshui ambient conditions on attitudes, social norms perceived behavioral control contribute to the positive relationship between social factors and individuals' expectations of the current literature of the tourism real estate industry, extending the TPB [6] by adding external aspects such as the government role, the infrastructure attribute, and the architecture style into the model. In particular, government policy can enhance customers' attitudes toward tourism real estate, therefore, playing an essential role in reducing the high pressure on legality when buying such housing [13]. The insignificant effects of government policy on social norm perceived behavioral control and purchase intention indicates that such policy only can influence an individual's attitude, who has specific purposes for their housing purchase rather than establishing a belief in a social group. Moreover, tourism real estate is a premium product in which customers must be qualified enough to participate in the market. By contrast, social infrastructure tends to the community rather than individual's aspects; hence, social infrastructure can positively influence social norm and create a feeling of behavioral control, especially for disabled people. This explains the insignificance of social infrastructure on customers' attitudes.

Moreover, Fengshui ambient conditions are a popular architectural design for Asian and Western cultures [5, 23]. Therefore, the positive relationship between Fengshui ambient conditions and the TPB's three core variables, including attitudes, subjective norm, and perceived behavioral control, are consistent with the philosophy of Sia et al. [27], in which housing purchase will be motivated by favorable Fengshui factors. As a result, a social norm based on the Fengshui principle was established in the tourism real estate market. Moreover, favorable Fengshui factors bring the feeling of fortune to customers, leading positive attitude toward the intention to purchase the housing.

### 5.2. Managerial implications

This paper also brings practical implications the management. In particular, although government policy does not influence social norms, it can enhance customers' attitudes toward real estate tourism. Therefore, it reduces the feeling of risks of legality when buying such housing. Consequently, real estate managers should take advantage of these policies whenever the government enacts favorable housing laws to encourage customers' demand and create positive attitudes toward purchasing tourism real estate. In addition, marketers should show customers a favorable market for tourism real estate based on a deep knowledge of current government policies and new release regulations, in which positive attitudes can motivate intention to purchase, and it would be a good opportunity for salespeople to bring it to actual purchases.

Besides, social norms were found out to be the most influential predictor of intention to purchase tourism real estate. Therefore, managers should focus on factors that can enhance social norms. Moreover, this paper provides two fundamental constructs that can improve social norms, including Fengshui ambient conditions and social infrastructure. For example, essential attributes of the social infrastructure consist of recreational/sporting facilities, cultural activities, distance to favorite places, and transportation conditions. In addition, key success aspects of Fengshui ambient conditions comprise of lighting, maintenance, safety, fresh air, green areas, directions of door, wind, or balcony, and thermal comfort. All these factors should be included in brochures when introducing real estate to customers to develop optimal marketing strategies.

## 5.3. Limitations and recommendations

Although this paper has brought some novel insights into both practice and literature, there are still inevitable limitations that should be applied with caution. First, the online survey makes it difficult to control the respondents' understandability of the questionnaire's content. Therefore, to reduce the bias from this limitation, this research cleaned the data before the data analysis by deleting invalid responses (e.g., less than 1 minute of completion or the same reactions for all questions). Moreover, this research only focused on the direct effects of social factors on TPB's constructs but did not examine the possible mediating effects of these factors on intention to buy through social norms, perceived behavioral control, and attitudes. Therefore, further research should take an additional step to examine this mediation in the tourism real estate purchase intention model. In addition, demographic variables are considered essential moderators in much previous marketing research but were not examined in this research. Therefore, further research could examine possible moderating effects of demographic variables such as gender, age, or education level, which can provide different results in the relationships between constructs in the tourism real estate model.

## Supporting information

**S1 Table. Feng Shui factors for the surrounding environment element [44–48].**
(DOCX)

**S1 Data.**
(CSV)

## Author Contributions

**Conceptualization:** Khan Van Ma, Zafar U. Ahmed.

**Data curation:** Khan Van Ma.

**Formal analysis:** Phuong V. Nguyen.

**Investigation:** Phuong V. Nguyen.

**Methodology:** Phuong V. Nguyen, Zafar U. Ahmed.

**Project administration:** Khan Van Ma.

**Writing – original draft:** Phuong V. Nguyen.

**Writing – review & editing:** Phuong V. Nguyen.

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
