## [Decision Letter · Decision Letter 0]

14 Nov 2022

PONE-D-22-27650The role of Government Policy, Social Infrastructure and Fengshui in Intending to Buy Tourism Real EstatePLOS ONE

Dear Dr. Nguyen,

Thank you for submitting your manuscript to PLOS ONE. After careful consideration, we feel that it has merit but does not fully meet PLOS ONE’s publication criteria as it currently stands. Therefore, we invite you to submit a revised version of the manuscript that addresses the points raised during the review process.

We look forward to receiving your revised manuscript.

Kind regards,

Dan-Cristian Dabija, PhD

Academic Editor

PLOS ONE

Journal Requirements:

2. PLOS ONE does not copy edit accepted manuscripts (https://journals.plos.org/plosone/s/criteria-for-publication#loc-5). To that effect, please ensure that your submission is free of typos and grammatical errors.

3. Thank you for including your ethics statement:  "N/A".   

(1) For studies reporting research involving human participants, PLOS ONE requires authors to confirm that this specific study was reviewed and approved by an institutional review board (ethics committee) before the study began. Please provide the specific name of the ethics committee/IRB that approved your study, or explain why you did not seek approval in this case.

(2) Please provide additional details regarding participant consent. In the ethics statement in the Methods and online submission information, please ensure that you have specified (1) whether consent was informed and (2) what type you obtained (for instance, written or verbal, and if verbal, how it was documented and witnessed). If your study included minors, state whether you obtained consent from parents or guardians. If the need for consent was waived by the ethics committee, please include this information.

**Additional Editor Comments:**

The manuscript has been reviewed by experts in the field. They suggested some major revisions. Please find attached their recommendations.

Furthermore, from my perspective, there are also some more recommendations which you should consider:

All abbreviations used in the manuscript should be highlighted as key words.Introduction.You should better explain how the paper adds value to the TPB.It should be explained how the research question is implemented in the paper, especially in the practical part.The last paragraph of the introduction should contain a brief description of the papers next sections.

Probably the lit review should start with explaining the TPB.

Please add more references and try to deduct the hypothesis based on more references.

Methodology

Please also explain the relevance of your research context for the international literature.Please also present the R2 and explain the prediction power of the model. I have not found any information about the VIF (method common bias) and the SRMR (goodness of fit of the estimated / saturated model). Please also include these aspects.Results.Please also include a table with the scales that you have used. Please include references for the constructs. Please also include the item loadings.

The paper is missing a clear section of discussions where own results are compared to previous findings. Please extend the paper. Here you can cite as many references as possible, thus pinpointing the originality of your paper.

Conclusions

Here no references should be cited.

Should consist from:

theoretical implicationsmanagerial contributionslimitationsfuture research perspectives

More references should be cited. References should be up to date.

You need to include a figure with assessment model, with all results!

Reviewers' comments:

Reviewer's Responses to Questions

**Comments to the Author**

1. Is the manuscript technically sound, and do the data support the conclusions?

Reviewer #1: Yes

Reviewer #2: Partly

2. Has the statistical analysis been performed appropriately and rigorously? 

Reviewer #1: Yes

Reviewer #2: Yes

3. Have the authors made all data underlying the findings in their manuscript fully available?

Reviewer #1: Yes

Reviewer #2: Yes

4. Is the manuscript presented in an intelligible fashion and written in standard English?

Reviewer #1: Yes

Reviewer #2: No

5. Review Comments to the Author

Reviewer #1: it is an intresting manuscript with sound methodology and well written. I have some suggestion for improvment:

why authors use TPB. why they added new variables to the theory? ّfurthermore i suggest authort start article as this formula:

why husing its important for tourist, what is the first step for iprove policy in this regard, which model can help, why tpb is good, why you need revised Tpb, How you can improve , why these new variables, Tpb. New variables, Hypo/s. methodology etc

furthermore for the why Tpb is important and why ew variables i suggest authors read these article:

How rationality, morality, and fear shape willingness to carry out organic crop cultivation: a case study of farmers in southwestern Iran

Investigating the effect of moral norm and self-identity on the intention toward water conservation among Iranian young adults

Application of the Theory of Planned Behaviour to predict Iranian students' intention to purchase organic food

Reviewer #2: Thank you for giving me an opportunity to review the paper entitled: “The role of Government Policy, Social Infrastructure and Fengshui in Intending to Buy Tourism Real Estate”.

There are several suggestions given for the improvement of the paper:

1. For abstract, authors should include and highlight the novelty of the study. Author should trim it down to not more than 250 words. At first, author could talk about the main purpose of the study and then novelty of the study. Then talk about the methodology such as sample and software used to analyse the model and hypotheses. After that, author could include the key findings and contribution.

2. Overall, I think the introduction is not well written. Any relevant past studies conducted before? If yes, authors could include it in the introduction. What make this study unique, and novel as compared to past studies? The importance of the study is not sufficiently explained. Authors did mention that past studies do not sufficiently test the internal process of this causal relationship. Author does highlight the issues. However, it relate it well to the context of the study and the gap of the study.

3. There is many missing information. The sequences of the introduction element are required to be reorganized. The introduction should be included as follows:

(1) Briefly describe and illustrate the current issue.,

(2) Why such study with proposed research gaps is important?,

(3) How this research gap relates to current issue?,

(4) Why such underexplored piece of work is important to be tested in your study?,

(5) Any similar studies conducted in the past?,

(6) What is the uniqueness of this study as compared with past empirical studies? and

(7) What are your research objectives?

(8) What are the contributions of the studies?

4. What is your underpinning theory? How this/these theories lay a support to your study and research model?

5. Why is Theory of planned behavior used in this study? Why not other behavioural intention theory?

6. All sections of literature review can be improved substantially. Authors should be reminded that a good literature review is NOT simply a list describing or summarizing several articles; a literature review is discursive prose which proceeds to a conclusion by reason or argument. However, a good literature review should show signs of synthesis and understanding of the topic. Thus, I urge authors to revise it substantially and also provide a table to demonstrate the past research findings.

7. Please include more recent citations (2019-2022). Please also check the citation style. Please ensure consistency and comply to journal requirement.

8. The proposed model should be explained as well by using the underpinning theory as a base.

9. Please clearly define the target respondents. Any criteria to select the respondents? How to consider them as respondents?

10. What was sampling technique used to select respondent? Convenience sampling? Why? How? Next question is how do you ensure the generalizability and representativeness of the sample toward the targeted population? Any selection criteria? How do you select the respondent for your study? Any procedure of selection? Please justify.

11. The procedure of data collection is not clear. Should provide more information about how authors collect the data, how to approach the respondent, how to identify them to participate in the survey? Try not to exaggerate it and the explanation should be more reasonable and logic.

12. Any pre-test and pilot test are conducted?

13. Any remedies to validate the adapted research instruments?

14. The research method should be only had two sub-sections: sample and procedure and research instruments.

15. Why PLS software used to analyse the data and test the model? Should provide explanation as well.

16. The measurement items description is very brief. Should provide more detail such as how many items from each variable and who is the one originally developed the scale.

17. It is good to include control variables as it may also influence the result of the study. The inclusion of control variables is required reasonable justification. Why these control variables included in this study?

18. Common method bias test should be conducted as this is a self-reported study.

19. The discussion and conclusion section structure should be revised as follows:

Discussion of key findings

Theoretical Implications

Practical/Managerial Implications

Limitations and Future Research

Conclusion

20. For discussion, authors need to ensure the key findings are discussed. The discussion section is where you delve into the meaning, importance and relevance of your results. It should focus on explaining and evaluating what you found, showing how it relates to your literature review and research questions, and making an argument in support of your overall conclusion, especially the mediation result.

21. For theoretical implications, it is too shallow. How do you imply these findings and compared with past study’s findings?

22. For practical/managerial implication. I would suggest author to provide implications based on the current practices and policies.

23. Please revisit the limitations as I found it is not adequately written. Suggest author to carefully identify potential weakness of this study and propose suggestion for future research. For instance, authors suggested that to include demographic variable as moderator. In fact, authors have collected demographical information of the sample, so it can be tested in this study instead of suggesting to collect this information and test it in the future study.

24. Should have a conclusion section.

6. PLOS authors have the option to publish the peer review history of their article (what does this mean?). If published, this will include your full peer review and any attached files.

Reviewer #1: **Yes: **Masoud Yazdanpanah

Reviewer #2: No

---

## [Author Response · Author response to Decision Letter 0]

29 Dec 2022

Response to Journal, the Editor, and Reviewers

Journal Requirements:

Authors’ responses: 

Thank you for the notice. We have checked the author’s guideline and follow it strictly. 

2. PLOS ONE does not copy edit accepted manuscripts 

(https://journals.plos.org/plosone/s/criteria-for-publication#loc-5).

 To that effect, please ensure that your submission is free of typos and grammatical errors.

Authors’ responses: 

Thank you for the comments. We have read the manuscript thoroughly and check typos and grammatical errors, hence, the revised one has been proofread.

3. Thank you for including your ethics statement: "N/A". 

(1) For studies reporting research involving human participants, PLOS ONE requires authors to confirm that this specific study was reviewed and approved by an institutional review board (ethics committee) before the study began. Please provide the specific name of the ethics committee/IRB that approved your study, or explain why you did not seek approval in this case.

Authors’ responses: 

The current study involved human participants and was reviewed and approved by the ethics committee of the Center For Public Administration, International University, Vietnam National University-Ho Chi Minh City, Vietnam. Moreover, the ethics committee ruled that no formal ethics approval was required in this study because it did not collect any medical information, there was no known risk involved, it did not intend to publish anyone’s personal information, and it did not collect data from underaged respondents. This research has been performed in accordance with the Declaration of Helsinki. Informed consent for participation was obtained from respondents who participated in the survey. The respondents who participated in the survey online (using the google form) or face-to-face survey were asked to read the ethical statement posted on the top of the document (There is no compensation for responding, nor is there any known risk. Participants were asked not to provide their names to ensure that all information would remain confidential. They are strictly voluntary and may refuse to participate at any time). The survey only proceeds if they agree. No data was collected from anyone under 18 years old.

(2) Please provide additional details regarding participant consent. In the ethics statement in the Methods and online submission information, please ensure that you have specified (1) whether consent was informed and (2) what type you obtained (for instance, written or verbal, and if verbal, how it was documented and witnessed). If your study included minors, state whether you obtained consent from parents or guardians. If the need for consent was waived by the ethics committee, please include this information.

Authors’ responses: 

The current study involved human participants and was reviewed and approved by the ethics committee of the Center For Public Administration, International University, Vietnam National University-Ho Chi Minh City, Vietnam. Moreover, the ethics committee ruled that no formal ethics approval was required in this study because it did not collect any medical information, there was no known risk involved, it did not intend to publish anyone’s personal information, and it did not collect data from underaged respondents. This research has been performed in accordance with the Declaration of Helsinki. Informed consent for participation was obtained from respondents who participated in the survey. The respondents who participated in the survey online (using the google form) or face-to-face survey were asked to read the ethical statement posted on the top of the document (There is no compensation for responding, nor is there any known risk. Participants were asked not to provide their names to ensure that all information would remain confidential. They are strictly voluntary and may refuse to participate at any time). The survey only proceeds if they agree. No data was collected from anyone under 18 years old.

Authors’ responses: 

We identified in the cover letter.

Additional Editor Comments:

1. From my perspective, there are also some more recommendations which you should consider:

a) All abbreviations used in the manuscript should be highlighted as key words.

Authors’ responses: 

Thank you for the comments! Some significant abbreviations were provided in the keyword list including Theory of planned behavior (TPB), Ho Chi Minh City - Vietnam (HCMC), and Partial Least Square Structural Equation Modelling (PLS-SEM). Other abbreviations are well-known indices in the statistics, so we omitted them including composite reliability (CR), average variance extracted (AVE), and Heterotrait-Monotrait Ratio (HTMT). (p.2)

b) Introduction: You should better explain how the paper adds value to the TPB.

Authors’ responses: 

Thank you for the suggestion. We have added justifications for adding values of this paper to the TPB by clarifying the importance of economic and situational factors (government policy, social infrastructure, and Fengshui conditions) in the tourism industry and the absence of these factor and the real estate industry in the TPB literature. (p.5)

“Several papers examined the factors that affect customers’ intention to buy tourism real estate, including the government policy, social infrastructure, and Fengshui conditions (Ibem 2015, Sia 2018, You 2012). However, there is still a lack of research on the connection between these factors and individual expectations in the TPB, which is the well-known philosophy in explaining individuals’ intention to engage in a behavior toward various aspects: health activities, environmental management, education, customer choice, psychology, etc. (Ammar et al. 2020, Archie et al. 2022, Sánchez-Medina et al. 2014, Xin et al. 2019, Wollast 2021). However, TPB is limited to predicting desired behaviors through attitudes, self-control, and subjective norms, it still does not take into account economic or situational factors that may affect an individual’s intention to implement a behavior. In addition, the tourism real estate is still unexplored in the literature of TPB, in which scholars have not paid much attention. Consequently, this paper extends to the philosophy of TPB by adding important factors in the real estate industry including government policy, social infrastructure, and Fengshui conditions.” 

c) It should be explained how the research question is implemented in the paper, especially in the practical part.

Authors’ responses: 

Thank you for the suggestion. We added the explanation on what is our research question and how we deal with it by providing the role of TPB, additional contextual factors, and the data analysis method. Moreover, general contributions for practitioners were listed to clarify how the results of this paper can answer the research question and reach the purpose of the study. (pp. 4 – 5)

“Therefore, to succeed in this market, real estate firms must understand their customers' expectations by frequently involving customer research in the company's strategy. Besides, in the post-pandemic in which commuting restrictions were released from most of countries around the word, international revenge travel is projected to be the future of the tourism industry (Zaman et al. 2021), opening a huge potential market for Vietnam investors in TRE. However, the question is which factors can drive investors’ intention to buy such properties to deal with distinct characteristics and legal risks in the emerging markets, such as Vietnam. To answer these questions, through an extensive literature related to the real estate purchase intention, the Theory of planned behavior (TPB) appears to be prominent in explaining customers’ intention behaviors in this context, which was selected and extended by adding contextual factors to develop a specific model on influential factors motivating the TRE purchase intention. Moreover, the Partial Least Square Structural Equation Modelling (PLS-SEM) was chosen to examine the relationships in the conceptual model and provides descriptive statistics to prove the validity and suitability of the data with the measurement and the structural model, providing novel insights to literature and practitioners in the real estate industry. Particularly, managers can identify which essential aspects of government policy to pay close attention to limit the potential risks in legality. Moreover, which necessary nearby facilities and critical conditions (e.g., interiors, security, utilities, and neighborhood) to drive customers’ positive attitude, social norms, and behavioral control, leading to intention to buy are discussed through the relationships between examining constructs for developing appropriate marketing strategies.”

d) The last paragraph of the introduction should contain a brief description of the papers next sections.

Authors’ responses: 

Thank you for the comment. We have added the organization of this paper to the last part of the introduction to describe the content of the following sections: literature review, methods, results and discussion, and conclusions and discussions as follows. (p.6)

“The rest of this paper is organized as follows. Section 2 presents the relevant literature on the relationships between constructs in the conceptual model and the theories applied as the foundation to develop the theoretical framework. Section 3 summarizes the methods for collecting data, developing the measurement, and analyzing the data; additionally, sample demographics are described. Section 4 illustrates the results of measurement and model assessment. Section 5 discusses and analyzes the research results, then provides contributions to both theory and practice. Finally, this section concludes the paper through limitations and recommendations for future research directions.”

2. Probably the lit review should start with explaining the TPB.

Authors’ responses: 

Thank you for the comment. We have moved the TPB sub-section to be the first part of the literature as the theoretical foundation to develop the conceptual model. (pp. 35 - 36)

3. Please add more references and try to deduct the hypothesis based on more references.

Authors’ responses: 

Thank you for the comments. We elaborated the arguments and justifications in each section of the literature review through additional references.

4. Methodology

a) Please also explain the relevance of your research context for international literature.

Authors’ responses: 

Thank you for the comment. We added the significance of this research context to the international literature by explaining the popularity of TRE research across countries and the shortage of the relevant research in Vietnam despite its great potentials in this industry. (p…)

“The TRE industry has recently emerged and are getting attention from worldwide scholars across 55 countries (Kabil et al. 2022). Despite the existence of the long-run and a bi-directional causal relationship between foreign direct investment in real estate sector and tourism (Fereidouni and Al-Mulal 2014) and the great potential of Vietnam tourism to the economy, there is little attention from scholars to this market toward the emergence of the TRE industry. Therefore, the respondents were chosen as employed people currently living and working in HCMC – one of the biggest city in Vietnam and nearby provinces with a large number of potential investors.”

b) Results: Please also present the R2 and explain the prediction power of the model. I have not found any information about the VIF (method common bias) and the SRMR (goodness of fit of the estimated / saturated model). Please also include these aspects.

Authors’ responses: 

Thank you for the comment. The R2 was presented in Figure 2. We added the description of the prediction power in the sub-section 4.3 (p…) and the explanation of VIF values in the sub-section 4.2 (p. …)

“Finally, Full Collin VIF values were calculated according to (65) to check the common method bias with a cutting point of 3.3. Particularly, the results show an acceptable FCVIF values for examining constructs including government policy (1.639), social infrastructure (2.819), Fengshui ambient conditions (2.631), attitudes (2.587), perceived behavioral control (2.038), social norm (2.819), and intention to buy (2.975). Therefore, the common method bias is not an issue when testing the relationships in the model.”

“The purpose of PLS-SEM is to predict rather than explanatory modeling. Therefore, efforts for developing model fit statistics or sacrificing predictive power to achieve better ‘fit’ could even be harmful and have proven highly problematic (68), which explains for a fair model fit values such as SMRM saturated model = 0.049 < 0.08 (69), but SMRM estimated model = 0.084 > 0.08. Instead, the coefficients of determination (R2) and path coefficients are alternative indicators to predict the endogenous constructs such as attitudes, social norms, perceived behavioral control, and intention to buy. Particularly, government policy, social infrastructure, and Fengshui ambient conditions can explain 33.1%, 23.1%, 33.5% of the total variance of attitudes, social norm, and perceived behavioral control, respectively, which explain 64.8% total variance of intention to buy. (see Figure 2)”

c) Please also include a table with the scales that you have used. Please include references for the constructs. Please also include the item loadings.

Authors’ responses: 

Thank you for the comments. The testing measurement was added in the Appendix C with items loadings and references for each construct as suggested. 

Appendix C. Questionnaire construction and latent variables.

Latent Variable Observed Items Mean SD Items loadings

Intention to buy

(Zhang, 2020) If I want to purchase of any property in the future, I will buy this kind of properties 0.844 0.021 0.845

 In the future, if others want to purchase of any property, I will recommend others to buy this kind of properties 0.901 0.011 0.899

 I would tell people the benefits of buying such kind of properties 0.898 0.010 0.897

 I hope I can own such kind of properties in the future 0.854 0.017 0.856

 Even if I have already purchased such kind of properties, I would still consider buying more. 0.832 0.020 0.835

Subjective norm (Zhang, 2020) My family supports me in buying such kind of properties 0.924 0.011 0.924

 My friend supports me in buying such kind of properties 0.961 0.006 0.961

 My colleague supports me in buying such kind of properties 0.954 0.007 0.954

Government policy (You, 2012) Integrity and enforceability 0.886 0.014 0.886

 Normative ex-ante administration governance and ex-post indemnity 0.909 0.012 0.908

 Regulations for false advertisements 0.799 0.032 0.800

 Regulations for real estate specifications 0.898 0.013 0.898

 Legal policy on real estate tax 0.910 0.011 0.910

 Legal transparency of authorities and investors 0.914 0.010 0.914

 Supervision and sanction of the authorities 0.909 0.012 0.909

Social infrastructure (Ibem, 2015) Recreational/sporting facilities in the estate 0.939 0.010 0.939

 Spaces and facilities for cultural activities 0.931 0.008 0.931

 Distance to other entertaining places and Central Area 0.902 0.017 0.902

 Transportation facilities 0.874 0.019 0.874

Fengshui, Ambient condition of interiors and adequacy of security, utilities and neighborhood facilities

(Sia, 2018, Ibem, 2015) Main door direction and view direction of the whole building 0.779 0.027 0.781

 Direction of the main door of the apartment 0.849 0.020 0.754

 Direction of the balcony of the apartment 0.860 0.017 0.786

 Wind direction 0.843 0.021 0.783

 Natural lighting 0.843 0.017 0.874

 Circulation of fresh air 0.842 0.015 0.861

 Level of thermal comfort 0.830 0.024 0.850

 Security measures in the residence 0.754 0.019 0.849

 Fire safety measures in the residence 0.753 0.030 0.839

 Open spaces and green areas in the estate 0.786 0.021 0.850

 Management and maintenance of facilities in the estate 0.783 0.026 0.861

 External lighting in the housing estate 0.873 0.016 0.844

 Playground for children in the estate 0.859 0.022 0.844

 Medical and healthcare facilities in the estate 0.848 0.020 0.842

 Traffic vehicles parking facilities 0.847 0.019 0.832

 Facilities of handicapped and social welfares 0.837 0.020 0.755

Attitude (Zhang, 2020) I think it is wise to buy this kind of properties 0.927 0.010 0.928

 I think it is safe to buy this kind of properties 0.941 0.007 0.941

 I think it is beneficial to buy this kind of properties 0.934 0.007 0.934

 I think it is laudable to buy this kind of properties 0.928 0.011 0.928

Perceived behavioral control (Zhang, 2020) Whether or not I buy this kind of properties is completely up to me, if finances permit 0.848 0.019 0.849

 I am able to judge whether this kind of properties is good or not 0.873 0.017 0.872

 I am confident that if I want, I can find this kind of properties on the market 0.909 0.014 0.910

 I can handle any (money, time, information related) difficulties associated with my buying decision 0.897 0.013 0.897

5. The paper is missing a clear section of discussions where own results are compared to previous findings. Please extend the paper. Here you can cite as many references as possible, thus pinpointing the originality of your paper.

Authors’ responses: 

Thank you for the comment. We have added more discussions in the section 5 as well as more references to compare our results with previous findings. The practical implications was provided with the current practice and policies in Vietnam to strengthen our conclusions and discussions. (pp. 21- 24)

6. Conclusions

Here no references should be cited.

Should consist from:

• theoretical implications

• managerial contributions

• limitations

• future research perspectives

Authors’ responses: 

Thank you for the comment. We added more references in each sub-section as suggested. 

7. More references should be cited. References should be up to date.

Authors’ responses: 

Thank you for the comment. We have added more recent references, especially from PLOS ONE.

Ammar, N., Aly, N. M., Folayan, M. O., Khader, Y., Virtanen, J. I., Al-Batayneh, O. B., ... & El Tantawi, M. (2020). Behavior change due to COVID-19 among dental academics—The theory of planned behavior: Stresses, worries, training, and pandemic severity. PloS one, 15(9), e0239961.

Xin, Z., Liang, M., Zhanyou, W., & Hua, X. (2019). Psychosocial factors influencing shared bicycle travel choices among Chinese: An application of theory planned behavior. PloS one, 14(1), e0210964.

Sánchez-Medina, A. J., Romero-Quintero, L., & Sosa-Cabrera, S. (2014). Environmental management in small and medium-sized companies: an analysis from the perspective of the theory of planned behavior. PloS one, 9(2), e88504.

Wollast, R., Schmitz, M., Bigot, A., & Luminet, O. (2021). The theory of planned behavior during the COVID-19 pandemic: A comparison of health behaviors between Belgian and French residents. PloS one, 16(11), e0258320.

Archie, T., Hayward, C. N., Yoshinobu, S., & Laursen, S. L. (2022). Investigating the linkage between professional development and mathematics instructors’ use of teaching practices using the theory of planned behavior. Plos one, 17(4), e0267097.

Nguyen, H. D., Dang, C. N., Le-Hoai, L., & Luu, Q. T. (2021). Exploratory analysis of legal risk causes in tourism real estate projects in emerging economies: empirical study from Vietnam. International Journal of Construction Management, 1-13.

Kabil, M., Abouelseoud, M., Alsubaie, F., Hassan, H. M., Varga, I., Csobán, K., & Dávid, L. D. (2022). Evolutionary Relationship between Tourism and Real Estate: Evidence and Research Trends. Sustainability, 14(16), 10177.

8. You need to include a figure with assessment model, with all results!

Authors’ responses: 

Thank you for the comment. Figure 2 was added as the results of the structural model testing 

Figure 2. The hypothesis testing result

Reviewers' comments:

Comments to the Author

Reviewer #1: it is an interesting manuscript with sound methodology and well written. I have some suggestion for improvement:

1. Why authors use TPB. Why they added new variables to the theory? 

Authors’ responses: 

Thank you for the suggestion. We added more justifications for the reason why use TPB as a base to develop the conceptual model in the introduction and literature review by emphasizing a lack of research on tourism real estate applying TPB and limitations of TPB in predicting behavioral intention through only three factors: attitudes, perceived behavioral control, and social norm. Moreover, the importance of situational and contextual factors such as government policy, social infrastructure, and Fengshui ambient condition has been mentioned in many previous studies in the tourism real estate industry. However, there is a lack of connection on whether these factors can be influential motivating customers to purchase. Therefore, based on an extensive literature review, we added these factors to the TPB for examining their relationships with behavioral intentions. 

“Due to the distinct characteristics and the complexity in legal risks of the TRE industry in Vietnam (Nguyen et al. 2021), customers are hesitant about investing in or buying these properties. Therefore, to succeed in this market, real estate firms must understand their customers' expectations by frequently involving customer research in the company's strategy. The research question is which factors can drive investors’ intention to buy such properties to deal with distinct characteristics and legal risks in the emerging markets, such as Vietnam. To answer these questions, through an extensive literature related to the real estate purchase intention, the Theory of planned behavior (TPB) appears to be prominent in explaining customers’ intention behaviors in this context, which was selected and extended by adding contextual factors to develop a specific model on influential factors motivating the TRE purchase intention.” (p.4)

“Several papers examined the factors that affect customers’ intention to buy TRE, including the government policy, social infrastructure, and Fengshui conditions (Ibem 2015, Sia 2018, You 2012). However, there is still a lack of research on the connection between these factors and individual expectations in the TPB, which is the well-known philosophy in explaining individuals’ intention to engage in a behavior toward various aspects: health activities, environmental management, education, customer choice, psychology, etc. (Ammar et al. 2020, Archie et al. 2022, Sánchez-Medina et al. 2014, Xin et al. 2019, Wollast 2021). However, TPB is limited to predicting desired behaviors through attitudes, self-control, and subjective norms, it still does not take into account economic or situational factors that may affect an individual’s intention to implement a behavior. In addition, the TRE is still unexplored in the literature of TPB, in which scholars have not paid much attention. Consequently, this paper extends to the philosophy of TPB by adding important factors in the real estate industry including government policy, social infrastructure, and Fengshui conditions.” (p.5)

2. Furthermore, I suggest author start article as this formula:

Why housing is important for tourist, what is the first step for improve policy in this regard, which model can help, why TPB is good, why you need revised TPB, How you can improve, why these new variables, TBP. New variables, Hypo/s. methodology etc.,

furthermore for the why TPB is important and why new variables. 

Authors’ responses: 

Thank you very much for the suggestions. We have restructured the introduction as well as the literature review section to be in more logical and convincing flow. Particularly, Vietnam is a potential location to invest in TRE due to an increasing demand in tourism after COVID-19 and commercial real estate is emerging as an economic phenomenon in an emerging market as Vietnam (Nguyen at al. 2021). The importance of Moreover, details on how we collect and analyze the data to get the most efficient and optimal results were briefly mentioned in the introduction. For hypotheses development, more references were also added to strengthening the relationships of three adding factors: government policy, social infrastructure, and Fengshui ambient conditions and the final outcome of intention to buy. (pp. 6 – 7) 

3. I suggest authors read these article:

How rationality, morality, and fear shape willingness to carry out organic crop cultivation: a case study of farmers in southwestern Iran.

Investigating the effect of moral norm and self-identity on the intention toward water conservation among Iranian young adults.

Application of the Theory of Planned Behaviour to predict Iranian students' intention to purchase organic food

Authors’ responses: 

Thank you very much for the suggestions. We have learned a lot from these suggested papers to elaborate our paper. 

Reviewer #2: 

 1. For abstract, authors should include and highlight the novelty of the study. Author should trim it down to not more than 250 words. At first, author could talk about the main purpose of the study and then novelty of the study. Then talk about the methodology such as sample and software used to analyse the model and hypotheses. After that, author could include the key findings and contribution.

Authors’ responses: 

Thank you for the comment. The abstract was reduced to 254 words. (p.1)

“Customers' demands are constantly changing, becoming more complicated with higher requirements. This market in Vietnam is still new and emerging and has encountered numerous issues regarding government policy, finance, and land authorization for constructing, owning, and managing. Because the form of tourism real estate is still new, customers are hesitant about investing in or buying these properties. However, there is still a lack of research on the connection between these factors and individual expectations in the well-known philosophy of the Theory of Planned Behavior (TPB), leading to behavioral intentions. Therefore, to fulfill the gap in the previous literature, this paper aims to investigate the connection between these factors with core variables of TPB, hence, addressing the current problems in the real estate industry. 471 valid respondents in Vietnam were collected for data analysis through two survey approaches. PLS-SEM was used to test hypotheses due to the relationship complication in the conceptual models. The results show that government policy influences attitudes and perceived behavioral control, whereas social infrastructure affects social norms and perceived behavioral control. Moreover, Fengshui ambient condition also positively influences all three core factors: attitudes, social norms, and perceived behavioral control. Finally, these factors impact on intention to buy tourism real estate. Through results, this paper has developed a purchase intention model through social aspects of the tourism real estate industry and demonstrates the connection between social factors and individuals’ expectations for a purchase intention. Thereby, recommendations of marketing strategies based on these findings were suggested to attain the optimal result for sales.”

2. Overall, I think the introduction is not well written. Any relevant past studies conducted before? If yes, authors could include it in the introduction. What make this study unique, and novel as compared to past studies? The importance of the study is not sufficiently explained. Authors did mention that past studies do not sufficiently test the internal process of this causal relationship. Author does highlight the issues. However, it relate it well to the context of the study and the gap of the study.

Authors’ responses: 

Thank you for the comments. We have elaborated the introduction by adding more justifications on the current issue in the TRE that requires close attention to develop a model for motivating customers’ purchase intention as what in this paper. We also added more arguments for the importance of applying TPB and adding factors to the TPB. (pp. 3 – 6)

3. There is many missing information. The sequences of the introduction element are required to be reorganized. The introduction should be included as follows:

(1) Briefly describe and illustrate the current issue.,

(2) Why such study with proposed research gaps is important?,

(3) How this research gap relates to current issue?,

(4) Why such underexplored piece of work is important to be tested in your study?,

(5) Any similar studies conducted in the past?,

(6) What is the uniqueness of this study as compared with past empirical studies? 

(7) What are your research objectives?

(8) What are the contributions of the studies?

Authors’ responses: 

Thank you for the comment. We restructured the introduction as suggested to have a better and more logical flow in explaining and introducing the paper. (pp. 3 – 6)

4. What is your underpinning theory? How this/these theories lay a support to your study and research model?

Authors’ responses: 

Thank you for the comment. TPB provided a strong foundation for the effect of attitudes, social norm, and perceived behavioral control on intention to buy. Therefore, these three factors contribute the mediating role in the model as a middle step between situational and contextual factors and final outcome of intention to buy. Particularly, The philosophy when using TPB is that independent factors including government policy, social infrastructure, and Fengshui conditions are external factors from the environment and society triggering internal factors of inside customers’ mind, which motivates behavioral intentions. (pp. 6-7)

5. Why is Theory of planned behavior used in this study? Why not other behavioral intention theory?

Authors’ responses: 

Thank you for the comment. We used TPB as the foundation to develop the model, as TPB is limited to predicting desired behaviors through attitudes, self-control, and subjective norms, it still does not take into account economic or situational factors that may affect an individual’s intention to implement a behavior. In addition, the TRE is still unexplored in the literature of TPB, in which scholars have not paid much attention. Consequently, this paper extends to the philosophy of TPB by adding important factors in the real estate industry including government policy, social infrastructure, and Fengshui conditions. 

“Several papers examined the factors that affect customers’ intention to buy TRE, including the government policy, social infrastructure, and Fengshui conditions (Ibem 2015, Sia 2018, You 2012). However, there is still a lack of research on the connection between these factors and individual expectations in the TPB, which is the well-known philosophy in explaining individuals’ intention to engage in a behavior toward various aspects: health activities, environmental management, education, customer choice, psychology, etc. (Ammar et al. 2020, Archie et al. 2022, Sánchez-Medina et al. 2014, Xin et al. 2019, Wollast 2021). However, TPB is limited to predicting desired behaviors through attitudes, self-control, and subjective norms, it still does not take into account economic or situational factors that may affect an individual’s intention to implement a behavior. In addition, the TRE is still unexplored in the literature of TPB, in which scholars have not paid much attention. Consequently, this paper extends to the philosophy of TPB by adding important factors in the real estate industry including government policy, social infrastructure, and Fengshui conditions (Appendix A).” (p.5)

“Although TPB was not applied much in the TRE research due to its new emergence, TPB has been widely applied in real estate studies, therefore, the effect of attitudes, social norms, and perceived behavioral control on behavioral intention to purchase the property has been consolidated across countries (Al-Nahdi et al. 2015, Islam et al. 2022, van Haaster-De Winter et al. 2022, Wu et al. 2021, Zhang et al. 2020), and so on. However, the findings are not consistent across studies. Particularly, there is no significant relationship between perceived behavioral control and intention to buy real estate in Saudi Arabia (Al-Nahdi et al. 2015), whereas subjective norms do not significantly affect behavioral intentions to purchase apartments in Bangladesh or real estate projects developed on industrial brownfields in China (Islam et al. 2022, Zhang et al. 2020). Besides, all three factors of customers’ perspective: attitudes, social norms, and perceived behavioral control positively influence intention to use nature-inclusive design and construction concepts (van Haaster-De Winter et al. 2022) or green residence purchase intention (Wu et al. 2021). Therefore, it is necessary to examine the relationship between these three factors and behavioral intention in the TRE sector, in which there is an absence of TPB application in determining antecedents of behavioral intentions.” (p.12)

6. All sections of literature review can be improved substantially. Authors should be reminded that a good literature review is NOT simply a list describing or summarizing several articles; a literature review is discursive prose which proceeds to a conclusion by reason or argument. However, a good literature review should show signs of synthesis and understanding of the topic. Thus, I urge authors to revise it substantially and also provide a table to demonstrate the past research findings.

Authors’ responses: 

Thank you for the comment. We have added more justifications in each section of the literature review to elaborate the arguments in this section. However, due to the word-limit, we only added main points to support our assumptions. Moreover, a table of the literature review on recent research on TRE was added in the Appendix A. (pp. 30-33)

Appendix A. Recent research on the TRE.

Study Purpose Main constructs Theory Key Findings

Barrantes-Reynolds, 2011 Implications and behavior of the expansion of TRE in Costa Rica N/A Case study TRE is not a form of tourism but a form of real estate business competing with holiday tourism and eco-tourism in the touristic market.

Hof & Blázquez-Salom, 2013 Determining the site-specific spatial, temporal and planning pathways by which activities and decisions of residential tourists and developers purchasing property in Majorca. N/A Case Study - Real Estate Tourism Shift.

- Transport Megaprojects Influence.

- Changes with the Crisis in the Land Use Regulatory Planning

Fereidouni, & Al-Mulali, 2014 Examining the connection between foreign direct investment in real estate sector and international tourism among OECD countries. Foreign direct investment,

International tourist departures,

International tourist arrivals,

International tourist the Granger causality test There is the existence of the long-run and a bi-directional causal relationship between foreign direct investment in real estate sector and international tourism.

Tsai et al., 2015 Examining determinants of TRE prices based on theme parks in China Price as the dependent variable

Others: area, floor, age, green ratio, decoration, landscape view,

Distance (to theme park, hotel, metro), number of bus station. Hedonic pricing Whereas distance to metro and the architectural features of the property significantly affect TRE value, distance to theme parks negatively affect price.

Liu et al., 2016 Investigating Chinese consumers' perceptions of brand personality of TRE companies and classifying consumers based on brand personality perceptions.

 Humanity,

Excitement,

Status enhancement,

Professionalism,

Wellness. Brand personality - Five brand personality factors: humanity, excitement, status enhancement, professionalism and wellness.

- Three consumer segments with distinct brand personality perceptions: status/ humanity consumers, wellness seekers and professionalism minders

Liu et al., 2019 Discovering the importance of brand personification in consuming TRE products in China - Personified brand personality dimensions: excitement, professionalism, status enhancement.

- Non-personified brand personality dimensions: humanity, wellness.

- Self-congruity,

- Brand personality Self-congruity theory Brand personality dimensions had larger effect on self-congruity than non-personified dimensions;

- Self-congruity mediates the relationship between brand personality dimensions and brand loyalty.

Nguyen et al., 2021 Examining legal risks in TRE projects and the relationship between customers’ characteristics and legal risk assessment in Vietnam. Legal problem factors,

Industry experience,

Project roles Legal risks - Participants’ industry experience and legal problem knowledge has a positive effect on legal risk cause assessment. 

- Project roles dis not moderate the effect of industry experience on legal problem knowledge with risk assessment

Ying, 2021 Proving that the real option method can enhance and optimize the investment decision-making on TRE in the US. Project value,

growth rate of GBM, fluctuation rate of GBM Case study,

Classic American Real Option Model - TRE investment is fully consistent with real option in the uncertain spatiotemporal attributes: uncertainty, irreversibility, and timeliness. 

- TRE project carries features of real option. 

Kabil et al., 2022 Providing an understanding of the link between the tourism and real estate sectors from the literature perspective. N/A N/A Providing how TRE is predominantly composed of practical research based on primary data and applied in different spatial units as case studies (e.g., coastal areas, cities and national and international units). - Providing a roadmap for the research streams of the TRE field.

This study Developing a model of TRE intention to buy in Vietnam Government policy, social infrastructure, Fengshui ambient conditions, attitudes, perceived behavioral control, social norm.

Intention to buy as the dependent variable. TPB Government policy, social infrastructure, Fengshui ambient conditions are three important determinant of attitudes, perceived behavioral control, social norm, driving intention to buy TRE.

Government policy does not affect intention to buy, perceived behavioral control, and social norm.

Fengshui ambient conditions is the only determinant affecting of attitudes, perceived behavioral control, and social norm.

7. Please include more recent citations (2019-2022). Please also check the citation style. Please ensure consistency and comply to journal requirement.

Authors’ responses: 

Thank you for the comment. We added recent citations through the manuscript, especially citations from PLOS ONE.

Ammar, N., Aly, N. M., Folayan, M. O., Khader, Y., Virtanen, J. I., Al-Batayneh, O. B., ... & El Tantawi, M. (2020). Behavior change due to COVID-19 among dental academics—The theory of planned behavior: Stresses, worries, training, and pandemic severity. PloS one, 15(9), e0239961.

Xin, Z., Liang, M., Zhanyou, W., & Hua, X. (2019). Psychosocial factors influencing shared bicycle travel choices among Chinese: An application of theory planned behavior. PloS one, 14(1), e0210964.

Wollast, R., Schmitz, M., Bigot, A., & Luminet, O. (2021). The theory of planned behavior during the COVID-19 pandemic: A comparison of health behaviors between Belgian and French residents. PloS one, 16(11), e0258320.

Archie, T., Hayward, C. N., Yoshinobu, S., & Laursen, S. L. (2022). Investigating the linkage between professional development and mathematics instructors’ use of teaching practices using the theory of planned behavior. Plos one, 17(4), e0267097.

Nguyen, H. D., Dang, C. N., Le-Hoai, L., & Luu, Q. T. (2021). Exploratory analysis of legal risk causes in tourism real estate projects in emerging economies: empirical study from Vietnam. International Journal of Construction Management, 1-13.

Kabil, M., Abouelseoud, M., Alsubaie, F., Hassan, H. M., Varga, I., Csobán, K., & Dávid, L. D. (2022). Evolutionary Relationship between Tourism and Real Estate: Evidence and Research Trends. Sustainability, 14(16), 10177.

8. The proposed model should be explained as well by using the underpinning theory as a base.

Authors’ responses: 

Thank you for the comment. We added more justifications in the sub-section of TPB to clarify the role of TPB in developing the model. 

“Through an extensive literature review in the TRE sector, there is no research examing determinants of intention to purchase TRE (see Appendix A). Most of them are qualitative research to explore new concepts or economic phenomenons (Nguyen et al. 2021), meta-analysis for a synthetic literature review (Kabil et al. 2022), case studies to assert an assumption as a typical example in the TRE (Barrantes-Reynolads, 2011; Hof & Blázquez-Salom, 2013; Ying 2021), or quantitative studies using secondary data to develop economic formulas (Fereidouni, & Al-Mulali, 2014; Tsai et al., 2015). Some studies on TRE use a questionnaire to collect primary data to emphasize the importance of brand personality or legal risks. Therefore, owing to the importance of TPB in the literature on behavioral intention, the absence of TPB in the TRE literature causes a significant gap in this topic. In addition, across studies applying TPB to various sectors, there is no research examining the direct effect of important factors such as government policy, social infrastructure, and Fengshui ambient condition on constructs in the TPB. However, these three factors play crucial roles in the real estate and the tourism industry (Ibem 2015, Sia 2018, You 2012), hence, for developing a conceptual model of customers’ TRE purchase intention, the inclusion of these three factors is significantly necessary. For this reason, government policy, social infrastructure, and Fengshui ambient conditions were added to TPB to extend the theory into the TRE field and based on the philosophy of TPB to develop hypotheses on the relationships between constructs in TPB and additional factors.” (p.7)

 “Several papers examined the factors that affect customers’ intention to buy TRE, including the government policy, social infrastructure, and Fengshui conditions (Ibem 2015, Sia 2018, You 2012). However, there is still a lack of research on the connection between these factors and individual expectations in the TPB, which is the well-known philosophy in explaining individuals’ intention to engage in a behavior toward various aspects: health activities, environmental management, education, customer choice, psychology, etc. (Ammar et al. 2020, Archie et al. 2022, Sánchez-Medina et al. 2014, Xin et al. 2019, Wollast 2021). However, TPB is limited to predicting desired behaviors through attitudes, self-control, and subjective norms, it still does not take into account economic or situational factors that may affect an individual’s intention to implement a behavior. In addition, the TRE is still unexplored in the literature of TPB, in which scholars have not paid much attention. Consequently, this paper extends to the philosophy of TPB by adding important factors in the real estate industry including government policy, social infrastructure, and Fengshui conditions (Appendix A).” (p.12)

9. Please clearly define the target respondents. Any criteria to select the respondents? How to consider them as respondents?

Authors’ responses: 

Thank you for the comment. We added details to define the target respondents who are at working age with a certain amount of income (company employees) or retired people. They are owning TRE or potential customers from the panel data from large real estate companies. Therefore, we asked for the permission to deliver the questionnaire to the panel date of existing and potential customers from five biggest real estate companies in Vietnam. (p.14)

“The respondents were chosen as employed people currently living and working in HCMC – one of the biggest cities, the economic capital, and the major destination for the TRE investment in Vietnam (Nguyen et al 2014) and nearby provinces with a large number of potential investors. The target respondents for this survey are people of working age with a certain amount of income (company employees) or retired people, who are owning TRE or potential customers from the panel data from large real estate companies: Novaland Group, Dat Xanh Group, Nam Long Investment Corporation, Hung Thinh Real Estate Business Investment Corporation, and FLC Group (Mordorintelligence 2021).”

10. What was sampling technique used to select respondent? Convenience sampling? Why? How? Next question is how do you ensure the generalizability and representativeness of the sample toward the targeted population? Any selection criteria? How do you select the respondent for your study? Any procedure of selection? Please justify.

Authors’ responses: 

Thank you for the comment. We applied the snowball sampling and purposive sampling approach to reach the target population because it can help to attain respondents who we were not aware even they are existing in the market and reach the precisely target population collection. The explanation for the generalizability and the representativeness of the sample was also added to clarify the procedure. (p. 14)

“Due to a lack of participants in the mass market, the snowball and purposive sampling were used to recruit the target respondents. These two approaches help to discover characteristics of the population that the authors were not aware existed in the market and reach expected characteristics of the target respondents (Levine 2014).” 

“Owing to the vast scale of these real estate companies, the questionnaire reached a large number of potential and existing customers in the real estate sector, achieving the generalizability and representativeness of the data, for example, the sample was evenly distributed across all levels of age and education.” (p. 15)

11. The procedure of data collection is not clear. Should provide more information about how authors collect the data, how to approach the respondent, how to identify them to participate in the survey? Try not to exaggerate it and the explanation should be more reasonable and logic.

Authors’ responses: 

Thank you for the comment. We added details to explain clearly the procedure of data collection. First, we conducted the translation-back method to ensure the content of the translated version. A survey link was created for the online survey to send to respondents through a google form with a screening question at the beginning of the questionnaire guarantee the accurate respondent collection. The definition of the tourism real estate was also provided to ensure the understandability of the participants to the research topic. Next, we conducted a pilot-test to check the initial measurement for last changes. The data collected was cleaned to eliminate invalid data or outliers. (p.14) 

12. Any pre-test and pilot test are conducted?

Authors’ responses: 

Thank you for the comment. We added details of the pre-test and pilot test procedure in the manuscript. (p.14)

“A translation-back technique was used in the pretest stage to ensure the understandability and accuracy of the translated content into Vietnamese with experts and scholars in the real estate field. After major changes in the translated version, a pilot test with 30 target respondents was conducted to check the reliability of the measurement for improving the questionnaire quality and accuracy.”

13. Any remedies to validate the adapted research instruments?

Authors’ responses: 

Thank you for the comment. For the pretest, experts and scholars in the real estate field were invited to evaluate the content of the translated version of the questionnaire to ensure the suitability of language use. For pilot test, Cronbach alpha was calculated to assess the reliability of the initial questionnaire, which was sent to 30 target respondents. Based on the comments and results of Cronbach alpha values, we made minor changes to the content of the questionnaire. For the final sample, after distributing to the mass, we cleaned the data to ensure the validity of the dataset based on three criteria: missing values, short time completion for online survey (less than 1 minute), and same answer for all questions.” (p.14)

14. The research method should be only had two sub-sections: sample and procedure and research instruments.

Authors’ responses: 

Thank you for the comment. We restructured the research method section with two sub-sections as suggested.

15. Why PLS software used to analyse the data and test the model? Should provide explanation as well.

Authors’ responses: 

Thank you for the comment. We added justifications for explaining the suitability of PLS-SEM and PLS software for data analysis in the assessment method sub-section: (1) the shortage of literature on the research model development toward the TRE, and (2) the advantages of the distribution assumption unavailability. (p.17)

“The data were analyzed using the reflection measurement model. Since the relationships among examined variables are complicated, PLS-SEM was applied by using SmartPLS software to test the research hypotheses, as it is the only software providing a full package of PLS-SEM analysis including a bootstrap procedure with a subsample of 5,000 to calculate standard deviation and create an approximate t-statistic value (Chin, 1998). The sample size was sufficient to guarantee the PLS method's regressions without singularities. PLS-SEM is a good option because the model is relatively sophisticated, but there is not much well-established literature on the conceptual model (Gefen 2000; Peng and Lai 2012). In addition, using PLS-SEM is unquestionably advantageous when considering the unavailability of distributional assumptions in several social science investigations (Hair et al. 2019). Therefore, the PLS path model was employed to estimate the measurement and structural models.”

16. The measurement items description is very brief. Should provide more detail such as how many items from each variable and who is the one originally developed the scale.

Authors’ responses: 

Thank you for the comment. We added the number of items and the original paper which developed these measures. (p.16)

“This study used the measure of previous studies with modifications to fit the TRE context. First, the Fengshui measure was adapted with minor revisions from the scale of Mak and Ng (2008) with 16 items, which derived from a study on architects’ perception (Mak and Ng 2005). Next, we modified four items adapted from You et al. (2012) and added three new items based on the current situation in Vietnam, including “Legal policy on real estate tax”, “Legal transparency of authorities and investors” and “Supervision and sanction of the authorities” with totally 7 items, which were originally developed from the legal regulation construct (Cooter and Ulen 2008). The measure of social infrastructure were modified from the scales of Ibem et al. (2015) with 4 items, originated from the concept of Ibem et al. (2012). In addition, four factors, including subjective norm (3 items), attitudes (4 items), perceived behavioral control (4 items), and intention to buy were measured by the scales of Zhang et al. (2020), which derived from the measurement of Ajzen (1991). The final questionnaire includes 43 items.

17. It is good to include control variables as it may also influence the result of the study. The inclusion of control variables is required reasonable justification. Why these control variables included in this study?

Authors’ responses: 

The research model includes two variables Age and Education level as control variables.

18. Common method bias test should be conducted as this is a self-reported study.

Authors’ responses: 

Thank you for the comment. We run the Full Collin VIF according to Knock (2005) to check the common method bias. (p.18). 

“Finally, Full Collin VIF values were calculated according to Knock (2005) to check the common method bias with a cutting point of 5 (Dodge 2008). Particularly, the results show an acceptable VIF values for examining constructs including government policy (1.639), social infrastructure (2.819), Fengshui ambient conditions (2.631), attitudes (2.587), perceived behavioral control (2.038), social norm (2.819), and intention to buy (2.975). Therefore, there were no multicollinearity issues when testing the relationships in the model.”

19. The discussion and conclusion section structure should be revised as follows:

Discussion of key findings

Theoretical Implications

Practical/Managerial Implications

Limitations and Future Research

Conclusion

Authors’ responses: 

Thank you for the suggestion. We restructured the section of discussion and conclusion (Section 5) as suggested. (pp. 21 – 24)

20. For discussion, authors need to ensure the key findings are discussed. The discussion section is where you delve into the meaning, importance and relevance of your results. It should focus on explaining and evaluating what you found, showing how it relates to your literature review and research questions, and making an argument in support of your overall conclusion, especially the mediation result. (p.21)

Authors’ responses: 

Thank you for the comment. We presents key findings of the paper in the key findings section by comparing the strength of the effect among independents to identify which determinant is the most or least important in contributing to the dependent, then they were discussed in the theoretical implication section for the relevance to previous studies and justifications for logic of the results as well as in the practical implication section for managers to develop suitable strategies to motivate customers’ purchase intention to buy the property. 

21. For theoretical implications, it is too shallow. How do you imply these findings and compared with past study’s findings?

Authors’ responses: 

Thank you for the comment. We added previous studies to compare with the result of this study. (pp. 21-23)

22. For practical/managerial implication. I would suggest author to provide implications based on the current practices and policies.

Authors’ responses: 

Thank you for the comment. The current practice and policies in Vietnam were added in the practical section including new law release, a potential market for foreign investors, and the current oversupply. (p. 23)

“According to HCMC Real Estate Association chairman, tourism real estate is oversupplied and lacking concrete regulations, resulting in unsustainable development (SGGP 2021). Therefore, finding a proper way to push customer purchase intention is necessary at this time. The results of this study can help managers to develop suitable strategies to trigger customers’ purchase intention to TRE based on external factors. In fact, in order to relieve financial burdens on the real estate market, the Ministry of Construction amended the 2014 Law on Housing and Real Estate to allow foreigners to own tourism real estate in Vietnam and foreign investors are extremely interested in the real estate market in Vietnam. Therefore, the TRE marketing in Vietnam is projected to compete fiercely in the upcoming years.”

23. Please revisit the limitations as I found it is not adequately written. Suggest author to carefully identify potential weakness of this study and propose suggestion for future research. For instance, authors suggested that to include demographic variable as moderator. In fact, authors have collected demographical information of the sample, so it can be tested in this study instead of suggesting to collect this information and test it in the future study.

Authors’ responses: 

Thank you for the comment. We added more details on limitations of this paper. We did not examine the moderating role of demographic variables in this paper, because the main purpose of this paper is to develop a conceptual model for identifying determinants of purchase intention to purchase TRE. In addition, due to the influential effects on relationships in the model of behavioral intentions, demographic variables are considered control variables. Therefore, it would be better to examine their moderation on relationships in the model of TRE intention to buy in detail in further research. (p.25)

“In addition, demographic variables are considered essential moderators in many previous studies on behavioral intention (Kekade et al. 2018, Park et al. 2021, Wu et al., 2015), but they were not examined in this research due to out of the main purpose of the study. Therefore, further research could examine possible moderating effects of demographic variables such as gender, age, or education level, which can provide different results in the relationships between constructs in the TRE model. Finally, no construction is free of risks (Latham 1994), hence, perceived risk should be considered a vital factor affecting customers’ perception and intention to purchase the property. Further research could assess the effect of various risk factors on the intention to purchase, especially risks related to the legality, such as owner-related, authority-related, contractor-related, and environment-related cause (Nguyen et al. 2021).”

24. Should have a conclusion section.

Authors’ responses: 

Thank you for the comment. We added the sub-section 5.4 as the conclusion of the study. (pp. 23-24)

“In summary, the findings of this study provided preliminary support to the TPB used as a foundation of the conceptual model for examining the effect of situational and contextual factors on intention to purchase TRE through customers’ perspectives such as norms, attitudes, and behavioral control. Although government policy does not significantly contribute to enhancing intentions to purchase, social norms, and behavioral control, it plays an important role in influencing customers’ attitudes toward TRE. Moreover, social infrastructure can simultaneously influence both customers’ attitudes and social norms. Especially, among the three independent factors, Fengshui ambient conditions seem the most essential factor when significantly contributing to developing three customers’ perspective factors. Interestingly, despite the important role in increasing behavioral intention in previous studies across industries and sectors, perceived behavioral control does not have a significant impact on the intention to buy TRE. This provides the distinct characteristics of TRE to the literature of tourism, real estate, and TPB. Through these findings, managers in the real estate industry can take advantage of government policies to trigger customers’ positive attitudes toward TRE. Therefore, continuously updating new legal regulations from the government is a success factor in the TRE. Furthermore, the findings also support the importance of architectural designs according to the Fengshui theory for TRE constructions. Finally, paying close attention to the current social infrastructure in surrounding areas of the TRE is another key point in successfully motivating customers’ perspectives toward intentions to purchase TRE.”

---

## [Decision Letter · Decision Letter 1]

24 Jan 2023

The role of Government Policy, Social Infrastructure and Fengshui in Intending to Buy Tourism Real Estate

PONE-D-22-27650R1

Dear Dr. Nguyen,

We’re pleased to inform you that your manuscript has been judged scientifically suitable for publication and will be formally accepted for publication once it meets all outstanding technical requirements.

Kind regards,

Dan-Cristian Dabija, PhD

Academic Editor

PLOS ONE

Additional Editor Comments (optional):

As the reviewers are happy with the revisions implemented by the authors, the paper can now be accepted.

Reviewers' comments:

Reviewer's Responses to Questions

**Comments to the Author**

1. If the authors have adequately addressed your comments raised in a previous round of review and you feel that this manuscript is now acceptable for publication, you may indicate that here to bypass the “Comments to the Author” section, enter your conflict of interest statement in the “Confidential to Editor” section, and submit your "Accept" recommendation.

Reviewer #1: All comments have been addressed

Reviewer #2: All comments have been addressed

2. Is the manuscript technically sound, and do the data support the conclusions?

Reviewer #1: Yes

Reviewer #2: Yes

3. Has the statistical analysis been performed appropriately and rigorously? 

Reviewer #1: Yes

Reviewer #2: Yes

4. Have the authors made all data underlying the findings in their manuscript fully available?

Reviewer #1: Yes

Reviewer #2: Yes

5. Is the manuscript presented in an intelligible fashion and written in standard English?

Reviewer #1: Yes

Reviewer #2: Yes

6. Review Comments to the Author

Reviewer #1: Dear Author thank you so much reagrding take care of my comments. I think you dont a great job and i suggest accepted in the curent form

Goodluck

Reviewer #2: I am happy to read the revised version of the paper. All my comments are well addressed by the authors. Thus, I have no further comments.

7. PLOS authors have the option to publish the peer review history of their article (what does this mean?). If published, this will include your full peer review and any attached files.

Reviewer #1: **Yes: **Masoud Yazdanpanah

Reviewer #2: No

---

## [Editor Report · Acceptance letter]

12 Apr 2023

PONE-D-22-27650R1 

The role of Government Policy, Social Infrastructure and Fengshui in Intending to Buy Tourism Real Estate 

Dear Dr. Nguyen:

I'm pleased to inform you that your manuscript has been deemed suitable for publication in PLOS ONE. Congratulations! Your manuscript is now with our production department. 

Kind regards, 

on behalf of

Professor Dan-Cristian Dabija 

Academic Editor

PLOS ONE